# Leptin Signaling in Obesity and Colorectal Cancer

**DOI:** 10.3390/ijms23094713

**Published:** 2022-04-24

**Authors:** Claudia Terezia Socol, Alexandra Chira, Maria Antonia Martinez-Sanchez, Maria Angeles Nuñez-Sanchez, Cristina Maria Maerescu, Daniel Mierlita, Alexandru Vasile Rusu, Antonio Jose Ruiz-Alcaraz, Monica Trif, Bruno Ramos-Molina

**Affiliations:** 1Department of Genetics, University of Oradea, 410048 Oradea, Romania; cristina_maerescu@yahoo.com; 22nd Medical Clinic, Department of Internal Medicine, “Iuliu Hatieganu” University of Medicine and Pharmacy, 400006 Cluj-Napoca, Romania; alexandra_rusu_22@yahoo.com; 3Obesity and Metabolism Laboratory, Biomedical Research Institute of Murcia (IMIB), 30120 Murcia, Spain; mariaantonia.martinez1@gmail.com (M.A.M.-S.); mangelesnunezsanchez@gmail.com (M.A.N.-S.); 4Department of Nutrition, University of Oradea, 410048 Oradea, Romania; dadi.mierlita@yahoo.com; 5Life Science Institute, University of Agricultural Sciences and Veterinary Medicine Cluj-Napoca, 400372 Cluj-Napoca, Romania; rusu_alexandru@hotmail.com; 6Faculty of Animal Science and Biotechnology, University of Agricultural Sciences and Veterinary Medicine Cluj-Napoca, 400372 Cluj-Napoca, Romania; 7Department of Biochemistry and Molecular B and Immunology, Faculty of Medicine, University of Murcia, 30100 Murcia, Spain; ajruiz@um.es; 8Department of Food Research, Centiv GmbH, 28857 Syke, Germany; monica_trif@hotmail.com

**Keywords:** *LEP*, *LEPR*, obesity, colorectal cancer, microbiome

## Abstract

Obesity and colorectal cancer (CRC) are among the leading diseases causing deaths in the world, showing a complex multifactorial pathology. Obesity is considered a risk factor in CRC development through inflammation, metabolic, and signaling processes. Leptin is one of the most important adipokines related to obesity and an important proinflammatory marker, mainly expressed in adipose tissue, with many genetic variation profiles, many related influencing factors, and various functions that have been ascribed but not yet fully understood and elucidated, the most important ones being related to energy metabolism, as well as endocrine and immune systems. Aberrant signaling and genetic variations of leptin are correlated with obesity and CRC, with the genetic causality showing both inherited and acquired events, in addition to lifestyle and environmental risk factors; these might also be related to specific pathogenic pathways at different time points. Moreover, mutation gain is a crucial factor enabling the genetic process of CRC. Currently, the inconsistent and insufficient data related to leptin’s relationship with obesity and CRC indicate the necessity of further related studies. This review summarizes the current knowledge on leptin genetics and its potential relationship with the main pathogenic pathways of obesity and CRC, in an attempt to understand the molecular mechanisms of these associations, in the context of inconsistent and contradictory data. The understanding of these mechanisms linking obesity and CRC could help to develop novel therapeutic targets and prevention strategies, resulting in a better prognosis and management of these diseases.

## 1. Obesity and Colorectal Cancer

Obesity is one of the leading disorders affecting humans worldwide; it is considered a global epidemic by the World Health Organization (WHO) [1], showing a complex profile of interactions and causality, including environmental, lifestyle, and genetic factors, and being involved in multiple disorders, including metabolic ones, with none of them completely understood [2]. Obesity is related to overweight, waist circumference, and visceral fat; in terms of overweight, the WHO obesity classification system grades obesity through body mass index (BMI) into five classes: normal weight (BMI: 18.5–24.9 kg/m^2^), overweight (BMI: 25.0–29.9 kg/m^2^), obesity (BMI: 30.0–34.9 kg/m^2^), severe obesity (BMI: 35.0–39.9 kg/m^2^), and morbid obesity (BMI ≥ 40 kg/m^2^) [3,4]. Obesity scoring is helpful for prevention, diagnosis, and treatment. Mutation events in the leptin gene (*LEP*) are associated with obesity. Annual weight loss of 5–10% is recommended by the Prevention and Therapy of Obesity guideline for people showing pathological overweight [5,6].

Research conducted in the fields of adiposity and obesity has led to its identification as a risk factor for several gastrointestinal cancers [7,8,9,10,11,12]. Since leptin level, signaling, and genetic variation are well known to be correlated with both obesity and colorectal cancer (CRC), obesity, which might be related to tumor molecular subtype, could be a potential interfering factor in CRC survival, indicating leptin as a potential marker for such purposes. Moreover, obesity is a risk factor in CRC development via inflammatory, metabolic, and endocrine processes, including cytokine signaling, all of which are linked to leptin’s role and could interfere in the recurrence of diseases and survival rates, considering the screening, prevention, and treatment of CRC [13]. Obesity determined through the body mass index has a significant correlation with higher values of leptin and an increased risk of CRC and death, whereas improved survival in the case of moderate adiposity may indicate a protective state for CRC [14]. However, there are inconsistencies related to tumor stage, treatment, and cancer-associated weight loss with respect to obesity classes [13,15]. Moderate and severe obesity shows a higher mortality risk among stage I–III CRC patients, but a potential prognostic side-effect of inflammation and obesity was reported in CRC metastatic cases [15,16]. High expression, rare mutations, and single-nucleotide polymorphisms (SNPs) of leptin were found in such obesity classes, delineating an association of leptin genetics, obesity, and CRC.

Over 90% of CRCs are adenocarcinomas (malignant neoplasms in the colon and rectum glandular epithelial cells); about 60–65% of CRCs are sporadic (in individuals with no CRC family history or inherited mutations related to CRC risk), through acquired somatic and epigenetic events, while about 35–40% of CRC are heritable through germline mutations. Nevertheless, it is important to emphasize that the heritable forms of CRC are not completely hereditary, with other risk factors also being determinants. However, CRC risk is increased in the case of CRC family history or other hereditary cancers. CRC presents four major stages, initiation, promotion, progression, and metastasis, occurring via three pathways, the adenoma–carcinoma sequence (in 85–90% of sporadic CRCs through adenoma precursor), the serrated pathway (in 10–15% of sporadic CRCs, involving hyperplastic polyps and sessile serrated adenomas), and the inflammatory pathway (<2% of all CRCs, involving chronic inflammation and dysplasia). Lifestyle, obesity, and CRC have many proven linked pathways that might reveal a direct relationship, despite specific traits related to countries and regional development grade. CRC has received international attention due to its rank as the third most commonly diagnosed cancer (10.0%) and the second leading cause of cancer deaths (9.4% of the total cancer deaths), with a higher incidence and age-standardized incidence or mortality rate (ASR) in countries with a higher Human Development Index (HDI) versus those with a lower HDI [17]. There is also a variation of incidence related to gender; in general, the ASR for all cancers is three times higher in men and two times higher in women in developed countries with high HDI (2.5 times overall), compared to less developed countries with a low HDI [10,11]. Despite the sex-related incidence of CRC being higher in men, the male-to-female ratio shows variations across regions, lifestyle factors, and diet [18].

Obesity is a pandemic and multifactorial disease featuring metabolic alterations, which can result in associated diseases, including those related to gut microbiome alteration. Furthermore, the gut microbiome interacts with host genetics, diet, and other environmental factors, also contributing to obesity and its related complications [19]. Obesity is involved in almost 15% to 20% of cancer cases [20]. In such terms, obesity is a condition associated with decreased microbial diversity, as well as taxon depletion, generated by complex interactions between diet and genetics, resulting in altered functionality and, thus, promoting dysbiosis, which is also related to obesity and CRC. Therefore, an altered microbiome could be a cause or a consequence of obesity [19]. It is well known that dietary habits and lifestyle vary across countries and regions; therefore, depending on their specific impact on human health, risk factors such as low-nutrient and high-calorie diets, low physical activity, rare hereditary diseases, age, gender, genetics, and ethnicity have been associated with obesity. CRC can also be considered a marker of socioeconomic development on the basis of its incidence-increasing trend in countries under transition, as well as in countries with a higher HDI [21]. Obesity also shows high prevalence rates in both developed and developing countries [22,23]. Lifestyle factors and diet, i.e., animal source-based food intake and a sedentary lifestyle, result in lower physical activity and a higher susceptibility to body weight gain and further obesity, which are independently associated with CRC risk, in addition to risk factors such as smoking, alcohol, and red or processed meat consumption [21]. Moreover, the consumption of whole grains, fiber, and dairy products is well known for its beneficial effects and influence on the gut microbiota, which is altered in CRC. These findings are in line with the fact that CRC is one of the clearest markers related to epidemiological and nutritional transition, showing incidence rates linked to Western lifestyles [24]. However, it could also be linked to multiple obesity forms, with a different causality and variation profile across countries and regions, contributed by host genetics and environmental factors. Thus, the association of obesity with CRC requires further investigation in larger and more complex studies.

The complex relationship between obesity and CRC may also be explained by interfering risk or protective etiological factors, which have positive associations with CRC, as well as a main influence on CRC. Increases in waist circumference, processed red meat intake, alcohol consumption, smoking, and a sedentary lifestyle were found to be associated with CRC risk, whereas physical activity and fiber/whole grain-based diets were found to be associated with a reduced risk of CRC (Figure 1). Obesity is definitely a convincing risk factor for CRC, with various studies indicating that waist circumference is a higher risk factor than BMI for CRC. On the other hand, recent studies have also linked obesity to visceral fat; therefore, CRC can be viewed as an obesity-related disorder, since adipocyte cell hypertrophy and excessive adipose tissue gain, mainly visceral adipose tissue (VAT), can result in pathogenic adipocyte and adipose tissue-related diseases, thus being an element of metabolic syndrome [25]. Moreover, adipose tissue is a key player in the innate immune response, contributing to systemic, chronic low-grade inflammation processes associated with visceral obesity, linked to tumor development through cytokine/adipokine secretion. Specifically, it mediates the inflammatory response in obesity via adipocyte-secreted molecules, i.e., hormones (the most relevant being adiponectin, leptin, resistin, and ghrelin), growth factors, and proinflammatory cytokines, interfering in CRC pathogenesis through cell growth, proliferation, angiogenesis, and expression processes. This results in the transformation of the normal colon mucosa in adenoma and adenocarcinoma [25]. The adipokine profile (e.g., leptin) could be a prognostic factor in CRC, whereby the adipokine or its analogues/antagonists could be useful in CRC management or chemoprevention.

Elucidation of genetic landmarks and their related mechanisms involved in obesity and its related disorders, including CRC, may result in novel target strategies for precision medicine adapted to specific obesity forms and cancer therapies. This review aims to summarize the current knowledge on the potential associations of leptin genetics with the main pathogenic pathways of obesity and CRC, in an attempt to understand the molecular mechanisms of these associations, in the context of inconsistent and contradictory data.

## 2. Leptin Signaling, Obesity, and Colorectal Cancer

Leptin is a multifunctional hormone with different roles, crucial to food intake, body weight control, and energy balance [26]. Leptin is encoded by the ob gene on the 7alpha31.3 chromosome in human, encoding a 167 amino-acid protein (16 kDa) whose first 21 amino acids function as a signal peptide and are cleaved before the circulating form of the protein (146 amino acids) is released in the blood flow [26,27]. Leptin shows a similar structure to other proteins of the cytokine family, belonging to the group of cytokines commonly called adipocytokines or adipokines [28]. As revealed by nuclear magnetic resonance, crystallized leptin is a four α helix protein (A–B–C–D), with a similar structure [26] and mechanism of action [29] to cytokines. All members of this family feature two long structures connecting the A and B helices, as well as the C and D helices. Leptin contains a unique disulfide linkage that connects two cysteines in the C and D helices, which seems to be critical for leptin’s structural integrity and stability [30]. Leptin’s link to its receptor is mediated by the A and C α-helix interface [31].

Leptin has been found in various tissues such as adipocytes, the stomach, and the gut [32,33,34,35,36,37,38,39,40], and it interferes with physiological processes such as energy metabolism, as well as the endocrine and immune systems. Leptin’s role as a hormone is linked to various endocrine functions, metabolism, and energy homeostasis, while its role as a cytokine is linked to inflammatory processes. High levels of leptin are found in the blood of obese patients. These factors all contribute to a decreased inflammatory state, resulting in susceptibility to various diseases, such as cardiovascular diseases, type 2 diabetes, and autoimmune diseases (Figure 2). Leptin is overexpressed in adipose tissue in cases of obesity, as well as due to resistance mechanisms leading to leptin’s inability to reach targeted cells, decreased leptin receptor (*LEPR*) expression, or altered signaling.

Leptin is one of the most important adipokines, generally considered as an “anti-obesity hormone” [41], although it displays various abnormal functions in obesity [42]. Data have indicated that leptin could play other roles in immune response, tumor invasion, and metastasis [41,42]. Adipocytes are involved in tumorigenesis, providing fatty acids, proinflammatory cytokines, and proteases [43,44]; since cancer cells utilize fatty acids, the involvement of adipocytes in tumorigenesis and metastasis is plausible [43]. Recent data have emphasized the brain’s ability to control the complex mechanism of food intake and storage [45]. Studies have indicated that specialized glial cells, tanycytes, which line the wall of the third ventricle in the median eminence, might be a key point for leptin entrance in the brain [46,47,48]. Researchers have proposed that the tanycytic *LEPRb–EGFR*-mediated transport of leptin might be highly important in lipid metabolism [45]. Leptin is associated with endocrine metabolism, as well as the regulation of appetite and energy expenditure; thus, decreased sensitivity to leptin may lead to metabolic disorders [49] and cancer, with its involvement demonstrated in many carcinogenesis-related signal pathways [50].

*LEPR* shows several spliced variants (i.e., *LEPRa*, *LEPRb*, *LEPRc*, *LEPRd*, *LEPRe*, and *LEPRf*) that are grouped into three classes (short, long, and secretive). Leptin binding to its receptor promotes various signal transduction pathways. After the binding process, the leptin receptor undergoes a conformational change, which is an important event for leptin signaling and activation of the associated JAK2, thus influencing the JAK–STAT pathway. Moreover, *LEPR* expression plays an essential role in colorectal carcinoma proliferation processes, whereby a lack of *LEPR* expression decreased tumor proliferation in most colorectal carcinoma cases, while high levels of expression resulted in neoangiogenesis and increased metastatic potential [51]. Leptin’s molecular ability to target its receptor has revealed its potential to enhance drug delivery, with leptin-derived peptides [52] showing the potential to decrease tumor growth in a CRC mouse model [53]. Serum leptin levels and leptin mutations are used in the diagnosis of congenital leptin deficiency and obesity [54]. The CRC microenvironment is related to *LEP* and *LEPR* expression, with significantly higher leptin levels in patients, indicating the potential of this molecular autocrine/paracrine signaling loop to interfere with tumor progression processes [55]. Leptin shows a mitogenic and antiapoptotic effect in CRC, which results in an enhanced invasiveness of colon cells and higher levels of expression related to tumorigenesis progress. Changes in *LEP* expression levels in the normal and altered mucosa of the colon (i.e., adenoma and adenocarcinoma) may indicate their potential influence in the multistep CRC carcinogenesis process. Therefore, unwanted *LEP* expression profiles, with a decreased anti-inflammatory and anti-cancerous effect, may be used in CRC prognosis, management, or prevention. Moreover, insulin resistance, hyperinsulinemia, hyperglycemia, oxidative stress, and leptin production underlie leptin’s potential role and associated mechanisms in obesity and CRC. Furthermore, alteration of the leptin/adiponectin balance can enhance JAK/STAT signaling in a similar way to inflammatory cytokines, which enables the remodeling of CRC cell signaling due to hormonal alterations, thus strengthening the link with CRC.

One study compared *LEP* and *LEPR* gene expression in healthy and CRC tissues retrieved from the public online IST database (Figure 3 and Figure 4) [56].

Altered expression of leptin and its receptor, which results in leptin resistance, is critical in obesity and its related complications. The association between obesity and cancers may also be partially supported by the high leptin level in blood circulation [57]; leptin’s involvement in the pathogenesis of various cancers, including CRC, has already been reported. As an adipokine, leptin seems to have a direct association with proliferation and apoptosis processes in CRC at various levels via the PI3K/AKT/mTOR signaling pathway, as well as with the improved expression of VEGF and VEGF-R2 via the PI3K, JAK2/STAT3, and ERK1/2 signaling pathways, as supported by in vitro, in vivo, and translational studies. Moreover, a review published in 2018 brought new insights into the role of the *LEP/LEPRb* axis in obesity-mediated cancers via the JAK/STAT, MAPK, and PI3K/AKT pathways, which are related to cell proliferation, cell migration and invasion, angiogenesis, vascular stimulation, and apoptosis. A high phosphorylation level of JAK/STAT signaling pathway molecules and transcriptional regulation of STAT3 downstream target molecules were reported in colorectal adenoma, whereas leptin-mediated proliferation and survival were related to the ObRL/STAT3 signaling pathway in colonic tumors; thus, tumor proliferation was inhibited in the case of deficiency of leptin and its receptor, even in severe obesity [50]. Moreover, obesity associated with high leptin levels can lead to chemotherapy resistance. In addition to the genetic leptin expression profile, risk factors such as high-energy diets, low consumption of health-promoting foods (i.e., fruits, vegetables, fiber), lifestyle, and age can alter adipose tissue functionality, all mediating the solid relationship between obesity and chronic inflammation processes in CRC. This highlights the value of adipocyte-secreted hormones in obesity and CRC prevention and management, through risk reduction and specific adapted therapies targeting chemoresistance prevention and recurrence [58].

## 3. Genetics of Leptin in Obesity and Colorectal Cancer

### 3.1. Background

Independently of adiposity level, large variation in the circulating level of leptin has been reported, of which 30–50% is presumably explained by genetic events. This validates the role of genetic variation in candidate genes in obesity, including leptin, thus motivating the interest in genetic variations in *LEP* and its receptor, as described below.

In addition to leptin’s main role in obesity, mostly referring to body weight regulation and energy homeostasis, it plays a major role in CRC by stimulating proliferation and inhibiting apoptosis [59]. Moreover, it is well known that leptin’s physiological mechanism of action is exerted throughout its receptor (*LEPR*), which is expressed in CRC [60], i.e., higher leptin serum levels [61]. Interestingly, the associations between *LEP* and *LEPR* gene variants and CRC are still contradictory [62,63], with mutation gain being a crucial factor enabling the genetic process of CRC [64].

Despite its role and mechanisms of action not yet being fully understood, leptin may represent the answer to obesity and its related pathologies, including CRC, as a function of its genetic interference. Genetic variations in the *LEP* gene are able to modulate its circulating levels and interfere with various pathophysiological processes. Obesity is linked to the expression, rare mutations, and polymorphic profiles of *LEP* and *LEPR* genes, as well as epigenetic events and other genes involved in energy regulation and metabolism. The influence of genetic and epigenetic factors on leptin expression associated with obesity has been investigated. since mutations, aberrant signaling, and epigenomic alterations have been described as molecular signatures in CRC. In general, *LEP* mutations may result in a lack of or smaller amounts of leptin, as well as its altered production (Figure 5), whereas exogenous administration of leptin is able to restore a normal health state in obesity. Despite studies indicating that genetic factors contribute to obesity and that no mutations in any one gene cause obesity in humans [65], several studies have reported extreme obesity due to a congenital deficiency in leptin synthesis. Various mutations of *LEP* have been associated with obesity grade and age of onset, varying across countries, regions, ethnicities, and other factors, while showing inconsistent associations with other obesity-related diseases. While obesity is a disorder with the most heritable profiles in inbred and outbred populations, 2–5% of cases of severe obesity are related to single-gene mutations, whereas 30% of cases of obesity are related to novel variants, mostly reported in highly consanguineous populations [66]. However, the data reported so far are limited. Moreover, of the 57 mutations identified in the *LEP* gene reported in humans, both homozygous and heterozygous, 1.5% are likely benign, 18% are benign, 1.5% are likely pathogenic, 21% are pathogenic, and 58% are unknown [67]; thus, the genotypic profile of *LEP* is of interest in obesity.

The inconsistent results of mutation screening in *LEP* may be due to the small number of studies performed across countries with a small number of cases. Thus, prevention screening strategies with larger assessments on healthy humans are required, along with complex association studies. Moreover, *LEP* and *LEPR* have shown main effects in CRC, with increased proliferation and metastasis [68], whereas decreases in metastasis and invasion have also been identified when controlling the expression of other genes [69]. Thus, further investigation is required.

In the next section, the main *LEP* and *LEPR* gene mutations associated with obesity and CRC are described to review their impact and potential link with obesity and CRC.

### 3.2. Rare Mutations in LEP and LEPR

Mutations in the *LEP* gene may result in different obesity forms, e.g., congenital leptin deficiency and extreme obesity, caused by a lack of leptin or an altered circulating form of leptin with no biological activity [70]. In general, rare mutations in *LEP* are associated with severe early obesity, through an effect on leptin signaling, leading to congenital leptin deficiency or leptin resistance. Moreover, rare loss-of-function mutations in the homozygous state of *LEP* lead to leptin deficiency, resulting in hyperphagia and severe obesity, whereas mutations in a heterozygous state result in partial leptin deficiency with higher body fat. On the other hand, the leptin serum level and *LEP* mutations are causal for obesity [54], being highly correlated with body fat mass and sequence variations in the leptin coding gene, which may affect the expression of leptin [71]. Additionally, several *LEP* and *LEPR* polymorphic profiles are associated with obesity, with only a small number of patients with early-onset extreme obesity being reported.

Only a few rare mutations in leptin described so far have been associated with obesity, showing a variation in profile across *LEP* gene sites and mutation type, as further outlined below. Following the first mutation in the *LEP* gene found in mice, a nonsense C→T mutation in codon 105, ob/ob deficient mice (*LEPob/LEPob*) became a common model for obesity. However, additional rare *LEP* mutations were reported in humans to be associated with obesity and its related comorbidities. This mouse nonsense *LEP* mutation changes an arginine into a stop codon, producing a truncated protein, which is further degraded in adipocyte cells, thereby causing a lack of leptin production and, thus, obesity [26,72]. The homozygous transversion (c.298G>T) modifies the active form of leptin into a biologically inactive protein, by changing an aspartic acid to a tyrosine (p.Asp100Tyr), such that a mutant protein is secreted, which does not bind or activate the leptin receptor; therefore, it is unable to decrease food intake and body weight [73]. Another biologically inactive form of leptin, caused by missense mutation p.Asn103Lys, has been linked to severe obesity [73]. A similar effect was observed for the missense recessive mutation in exon 3 of *LEP* gene c.350G>A (p.C117Y), leading to impaired protein function associated with severe obesity [74]. Furthermore, severe obesity was found in congenitally leptin-deficient humans of Pakistani origin caused by a homozygous frameshift mutation consisting of a single guanine deletion in codon 133 (Δ133G or g.13374delG) of the *LEP* gene, affecting the protein p.Gly133Valfs*15. This was the first genetic evidence of leptin as a regulator of energy balance [70]. Two rare mutations at codons F17L and V110M have also been associated with juvenile-onset obesity [75]. A homozygous missense substitution mutation C→T (c.313 C>T) in codon 105 of the *LEP* gene producing an Arg→Trp replacement (p.R105W) in the mature protein was reported in a family of Turkish origin [76], while a missense mutation g.15409C>G (p.S141C) was found in Turkmenistan [77]. The same ΔG133 homozygous frameshift mutation was also reported in a Pakistani consanguineous family with obese morbidity and congenital leptin deficiency [78]. Homozygous missense mutations g.13285C>A and c.309C>A in the *LEP* gene (p.N103K) have been found in an obese Egyptian patient [79], as well as in German [73] and Pakistani [2] cohorts. The homozygous missense transition g.13289C>T, (TTA to TCA) in exon 3 of the *LEP* gene leading to a p.L72S protein replacement was shown to be associated with mild obesity in Australia, whereby the mutated leptin was expressed but not secreted into blood circulation [80]. The leptin mutations c.481_482delCT, c.104_106delTCA, and previously reported frameshift c.398delG were reported in obese Pakistani patients [81]. A leptin homozygous frameshift mutation, G133_VfsX14, was associated with severe obesity in Pakistani consanguineous families [82]. Furthermore, a leptin mutation was identified in a consanguineous Indian family [83]. A nonsynonymous mutation H118L in exon 3 was associated severe obesity in Chinese [84]. A homozygous missense mutation in exon 3 of *LEP* gene c.298G>A (p.Asp100Asn), resulting in asparagine substituting aspartic acid at codon 100, was associated with severe obesity in India [85]. A homozygous missense mutation C350G>T (p.C117F) in exon 3 of the leptin-coding region was associated with severe obesity in two consanguineous Colombian sisters, showing evidence of the presence of monogenic leptin deficiencies in North and South America, in addition to those previously reported on other continents [86]. The current data on *LEP* mutations indicate a quite different genetic profile across countries; however, only a few studies have been performed, and a common association with a severe obesity state cannot be excluded. Consequently, larger screening studies can facilitate the discovery of new rare mutations possibly associated with the complex interaction mechanisms of obesity.

Only a few studies have reported mutations in *LEPR* associated with obesity, although early-onset obesity involving *LEPR* mutations is less prevalent. In general, it is associated with normal or high leptin values in plasma, nonmetabolic symptoms (altered immune system), or other disorders. In monogenic obesity, mutations have been mainly identified in *LEP* or its receptors mostly at the hypothalamus level. *LEPR* has six isoforms (A–F), which, along with leptin, are associated with obesity, with the first *LEPR* mutation being reported in North Africa (c.2598 + 1G>A). *LEP* and *LEPR* gene mutation signatures involved in colon cancer are mainly missense, frameshift, silent, nonsense, and splice site mutations (*LEP*: g8r, p23l; *LEPR*: x14_splice d124y, g179afs*35 v198a, s256y s303s, x429_splice l1094f, v430i s541p, w558* r573h, s595s l598p, v606v r612c, i814l i845t, v846v d921y, s927s q1034h, k1074t s1090r) [87]. Other studies have indicated that the allelic frequencies of *LEP* (A19G) and *LEPR* (Q223R, K109R, and K656N) SNPs related to obesity show ethnic variation; however, the associations were inconsistent, likely due to the complex pathogenesis of obesity, mediated by many genetic and environmental factors [88].

The above-described mutations identified in leptin might indicate an association with obesity grade and the age of onset. Furthermore, such mutations or new ones in the *LEP* gene may not yet be unraveled due to poor screening; thus, further assessment of *LEP* and *LEPR* could reveal new damaging mutations in these genes, useful for clinical testing, prevention, or treatment of various forms of obesity, as well as a better stratification of CRC patients.

### 3.3. Relevant SNPs in LEP and LEPR 

Despite there being over one million SNPs of the human genome, only a few specific SNPs, including those of *LEP*, have shown the potential to be associated with various conditions, including obesity and cancer [89]. Moreover, various polymorphisms in the *LEP* gene are linked to extreme obesity, such as D7S514, D7S680, D7S530 [90], D7S504, and D7S1875 [91], while the most frequently studied LEP SNPs are rs7799039 [92], rs2167270 [93], rs4731426, rs2071045 [94], and rs17151919 [95] as determinants in obesity [96]. The polymorphism C(−188)A (rs791620)in the promoter region of the *LEP* gene was associated with lean and morbidly obese subjects [97]. A recent colorectal cancer case–control study related to polymorphisms revealed the association of *LEPR* rs6588147 and rs1137101, as well as *LEP* rs2167270, with lower cancer risk, whereas *LEPR* rs1137100 was associated with cancer susceptibility [98]. Hence, only a relatively small number of *LEP* polymorphisms have so far been associated with obesity. Differences described in terms of altered sites and targeted nucleotides require further studies to establish a genetic profile underlying the influence of *LEP* SNPs on obesity. Moreover, *LEP* polymorphisms described so far have shown variations related to the obesity form or grade, which may be the result of other influencing factors such as clinical features.

On the other hand, of the 44 SNPs related to CRC risk, some of them overlap with leptin. Regarding the relationship of obesity with CRC, *LEP* polymorphisms such as rs7799039 and *LEPR* rs1137101 were shown to be associated with CRC risk, with specific alleles or genotypes linked to increased obesity risk [99]. The *LEP* A19G polymorphism (rs2167270) consisting of an A→G transition at nucleotide 19 in the exon 1 5′-untranslated region (Hager et al., 1998) was found to be associated with obesity in various cases [92,100,101], whereas a link was not found in [102]. A common *LEP* G2548A polymorphism (rs7799039) consisting of a G→A transition at nucleotide −2548 upstream of the ATG start codon in the 5′ promoter was also associated with obesity or body weight gain in various cases [93,100,103,104,105], whereas other reports failed to prove such an association [106,107,108,109]. *LEP* SNP rs7799039 was also found to be associated with changes in the lipid profile [110]. The contradictory effects of these SNPs could be explained by the limited number of available studies and the influence of other factors related to obesity not yet known or addressed, which should be considered in future complex studies.

Among the various genetic variants of the *LEP* gene, the G19A polymorphism has been investigated cancer risk susceptibility due to its beneficial effects related to cancer risk, tumor size, and metastasis [111]. Contradictory results have revealed the influence of this SNP to range from no association with CRC susceptibility in Caucasians from the USA to an association with lower colon cancer risk in Mexican patients [63,112]. The G19A *LEP* SNP may interfere with various processes such as RNA transcription, translation, and steadiness, thereby changing leptin protein expression and correlating with cancer risk [112]. Furthermore, rs2167270 (G19A) and rs7799039 (G2548A) can be considered relevant markers for obesity depending on ethnicity, with a significant genotype variation in obesity-related phenotypes such as subcutaneous fat reported [102]. Conflicting findings regarding the potential effect of the G2548A *LEP* polymorphism on leptin expression have also been reported [106]. Thus, the influence of these two common variants of *LEP* (A19G and G2548A) on obesity is inconclusive, limited by the insufficient number of available studies addressing such associations, as well as circulating levels of leptin.

Recently, the rs12535708 polymorphism in the intronic region of leptin was ranked in the top 10 contributing SNPs predicting child overweight/obesity, and it could represent an early biomarker for dysbiotic-prone obesity-associated microbiota, in addition to to other molecules involved in leptin-related mechanisms [89]. Additionally, *LEPR* rs12037879 might be associated with a marginal increase in CRC risk, influenced by smoking, cancer inheritance in the family, and *LEPR* rs6690625 [63]. Most LEPR SNPs have been assessed as potential modifying factors of the response to diet or of survival in cancer patients according to their BMI [88], with associations between these polymorphisms and obesity and CRC being contradictory or inconsistent.

Leptin association studies are restricted to the variants identified at the time of investigation. However, further insights revealing new genetic loci associated with obesity, influencing the circulating levels of leptin mediated or not by BMI are required. This can help in comprehensively clarifying the strong leptin–adiposity correlation, including other interfering factors.

### 3.4. Impact of Leptin Genetic Variation on Obesity and Colorectal Cancer

Obesity is considered one of the main environmental risk factors for CRC pathogenesis, with leptin being an important adipokine involved in both obesity and CRC. Insights into the genetics of leptin, independent of lifestyle and environmental factors, have indicated that all *LEP* mutations reported until now exhibit specificity related to population, rather than ethnicity [66]. This highlights the necessity of mutation screening in various populations, compiling the specific associated risk factors. Furthermore, even though the effects of *LEP* genetic variation and expression are well known in obesity, conflicting results have been found for normal leptin levels in blood circulation related to BMI and fat mass [113].

Considering the genetic origin of obesity, the high incidence of a severe obesity state related to novel and rare monogenic mutations, mostly found in highly consanguineous populations, could be explained by the practice of consanguineous marriages still occurring in many countries of the Middle East, Africa, and Asia, as well as isolated geographical regions, based on religious concepts, education, or specific traditions [66,74]. The occurrence of rare mutations may be linked to such habits. Despite the specific events described in consanguinity, such as the high levels of homozygosity and genomic loss-of-function mutations, which are able to inactivate or alter the function of genes, the assessment of consanguineous populations might be of interest for gene function characterization, including *LEP*, especially in consanguineous offspring, targeting the effects of such mutations. Therefore, there is a compelling necessity of systematically advanced genetic screening, not only in such countries, but also in countries where such assessments have not yet been undertaken, related to obesity and CRC association [66]. Common obesity, which is most prevalent, consists of a polygenic form due to multiple susceptibility loci resulting in a modest effect on BMI [114], whereas monogenic obesity due to mutations in a single gene is rare and has a large effect on severe adiposity in early life [66]. In terms of race and ethnicity, in Asians, visceral obesity, which is more prevalent compared to that reported in Caucasians, independent of BMI values, is associated with high CRC risk, especially in men, but also women if other factors linked to CRC risk, such as genetically determined BMI, lead to early-life adiposity. On the other hand, the anticancer effect of endogenous estrogens in the later life stage of women has been noted. CRC generally shows a genetic predisposition of low penetrance, thus explaining that an important percentage of CRCs clustered in families may be explained by acquired somatic mutations. This emphasizes the influence of other risk factors, with a similar cumulative risk profile to that described in obesity. Despite the low penetrance of heritable genetic variation in CRC and obesity, both acquired and inherited variation may be important factors, which may act together or separately on the basis of onset, age, environment, lifestyle, nutrition, sex, population, race, and ethnicity. In this line, single-gene mutations including those of *LEP* and *LEPR* genes are responsible for causing early-onset obesity [78], whereas genetic, genomic, and epigenetic alterations have been reported in syndromic forms of obesity, despite obesity typically being a multifactorial disorder showing a high heritability profile (50–75%), which is probably higher in early-onset cases [115]. However, mathematical models combining the effect of all common autosomal SNPs are only able to explain about 17% of the variance in body mass index [116].

Interestingly, although *LEP* heterozygosity was associated with higher metabolic efficiency and longevity in mice [117], such an effect was not assigned in humans, despite patients with *LEP* mutations showing the expression of more severe phenotypes of obesity. Patients with leptin deficiency showing early-onset severe obesity and mutated mice (ob/ob, db/db) are indistinguishable [66].

*LEP* gene mutations could reveal essential insights into obesity and its associated disorders; polygenic and monogenic leptin genetics could contribute to elucidating the mechanism associated with various cancer types, including CRC, with substantial gaps in the knowledge despite being in the third decade of research on leptin [66]. Rare loss-of-function mutations in *LEP* and its receptors involve common mechanisms related to altered pathways regulating leptin synthesis, whereby *LEP* level is correlated with *LEP* gene transcription, adipocyte size, and lipid content, independent of leptin resistance mechanisms, which are not clearly understood.

The potential link between obesity and CRC might be explained by metabolic changes able to drive to genetic and epigenetic alterations promoting tumor development. Metabolites and metabolic byproducts, such as ROS, are able to generate genetic lesions that may result in the genetic instability of chromosomes. It is known that excess ROS results in an increased mutation rate, thus favoring or maintaining oncogenic processes. Referring to the carcinogenic pathways of CRC, it is possible that different genetic profiles of leptin might be found according to specific pathways at different time points, considering the distinct and specific molecular signatures identified for the main oncogenes (TP53, APC, and BRAF) in the three CRC pathways and their proven relationship with *LEP*. In an attempt to further explain such considerations, each pathway is described below in the context of leptin. The classic CRC pathway, i.e., the adenoma–carcinoma pathway, accounting for most cases of sporadic CRC, is mainly associated with the development of the chromosomal instability-positive subtype, characterized by the gradual accumulation of genetic and epigenetic alterations, thereby driving the transformation of normal colon cells to cancerous cells. The serrated carcinogenic pathway of CRC, accounting for a smaller percentage of sporadic CRCs, shows distinct molecular profiles consisting of early mutation events leading to the uncontrolled proliferation of colon cells and a high CpG island methylation phenotype. In the third CRC pathway, i.e., the inflammatory pathway, dysplasia results from chronic inflammation in the flat mucosa with multifocality, thus obscuring lesion identification, in contrast to the discrete dysplastic precursor lesions found in the other two pathways of CRC. Likewise, in the inflammatory pathway, genetic events occur at different frequencies and time points, which could be linked to the role of leptin in inflammatory processes and its genetic background.

Comprehensive screening for *LEP* and *LEPR* mutations should be carried out to identify new variants, which can be integrated into current medical strategies. Recently, new treatment strategies for leptin pathways have emerged, showing the potential to modify these rare forms of obesity, where there is no causal therapy available [113].

In the past few years, new mutations in the *LEP* gene related to obesity and its associated disorders have been identified, which may be important for enabling progress toward understanding the physiopathology of obesity and its specific management [113]. Therefore, considering the success in obesity management based on congenital or acquired *LEP* mutations, progress toward elucidating the mechanisms linking obesity and CRC may be a potential direction for compiling new strategies and tools in medicine.

The inconsistent or paradoxical results assigned to leptin genetics and various cancer types, including CRC, could reside in interfering factors such as the genetic heterogeneity of cancers, malignancy grade, clinical heterogeneity, and exposure [118]. In addition, the discrepancies in the association of leptin genetics, obesity, and CRC may be explained by the insufficient sample size, statistics, genotyped markers, allele frequencies in various populations, linkage disequilibrium, or diet and lifestyle factors [99], emphasizing the need to assess *LEP* and *LEPR* gene variants in larger studies, including both obese and lean subjects, with epidemiologic data on dietary habits in different ethnicities. These factors are also linked with complex etiologic factors resulting from incompletely characterized interactions of genes, environment, and lifestyle [119].

Leptin genetics based on molecular background could be of interest in determining genetic susceptibility profiles related to obesity and CRC in terms of prevention, prognostic treatment, or management strategies. Furthermore, treatment options can be expensive; thus, the cultural and socioeconomic context should be considered, as the health system in many countries cannot provide early screening, diagnosis and treatment alternatives. Thus, timely screened genetic profiles could represent an efficient approach in the management of obesity and CRC. However, reality should be faced, considering that many countries still lack genetic screening facilities; therefore, the management of specific diseases, including obesity and CRC, is in an initial phase of using such advanced techniques. While some progress has been registered through financial investments in laboratory equipment, supportive actions are more than welcomed.

## 4. Interplay of Obesity, Gut Microbiota, and Leptin Signaling in Colorectal Cancer

The gut microbiota was recently considered as a key factor that can contribute to both initiation and development of CRC [120]. Recent studies have reported that CRC patients display significant alterations in specific bacterial groups with a potential impact on the mucosal immune response with respect to healthy controls [121,122,123]. This imbalance in intestinal homeostasis (dysbiosis) is mainly characterized by an enrichment in proinflammatory opportunistic pathogens, which could promote tumor formation [124,125]. Notably, patients with colorectal tumors at an early stage (advanced adenoma) have a different microbiota composition compared with those with advanced-stage tumors (definitive CRC) [121], suggesting that the gut microbiota could also participate in tumor progression.

On the other hand, emerging evidence suggests that the relationship between obesity and CRC could be mediated by alterations in the gut microbiota. Thus, several studies have shown that specific bacterial taxa linked to obesity could play a role in the etiology of CRC [120,126]. Interestingly, a recent study demonstrated that CRC patients with obesity display a specific microbiota profile mainly characterized by a reduction in butyrate-producing bacteria and an overabundance of opportunistic pathogens in comparison to nonobese CRC patients, suggesting that these changes were partially responsible for the higher levels of proinflammatory cytokines and gut permeability in these patients [123,127,128]. Chronic low-grade inflammation is a hallmark of both obesity and CRC etiology [13,129]; leptin itself has already been proven to be an inflammation inducer [130], along with other inflammatory cytokines. More recent data [131] have suggested that leptin enhances the activity of inflammasomes. Pham et al. reviewed the mechanism and roles of leptin in inflammasomes activation [131]. Furthermore, data are being accumulated regarding obesity and neurodegenerative diseases with respect to neuroinflammation [132]. A recently published review showed that obesity has not only peripheral but also central effects, in which aberrant microglial inflammation is a hallmark [132]. Remarkably, intestinal inflammation has been proposed as one of the possible mechanisms via which microbial dysbiosis may play a role in CRC carcinogenesis [133]. On the other hand, several studies have shown that the gut microbiota is a key factor that drives inflammation in the colon, and this inflammatory environment could be related to CRC development [120]. Moreover, increased gut barrier permeability in patients with obesity could have an additional impact on CRC development by inducing a metabolic endotoxemia that further increases tumor-promoting systemic inflammation and contributes to the production of proinflammatory cytokines [123,134].

The absence of leptin signaling in both leptin-deficient ob/ob and leptin receptor (*LEPRb*)-null *db/db* mice results in alterations in the composition of the gut microbiota (increased Firmicutes to Bacteroidetes ratio) [135,136]. It remains unclear, however, whether these changes in gut microbiome composition are due to the altered leptin action resulting from hyperphagia, from physiological changes associated with obesity, or from other leptin actions independent of food intake and adiposity. Interestingly, alterations in leptin signaling caused by a mutation in the extracellular domain of *LEPRb* have been associated with increased susceptibility to *Entamoeba histolytica* [137,138], suggesting a role of leptin signaling in the intestinal epithelium in the defense against gut pathogens. Additionally, it has been proposed that leptin signaling might potentially modulate bacterial populations within the gut independently of food intake, by controlling the expression of gut antimicrobial peptides [136].

On the other hand, it has been demonstrated that the gut microbiota is able to control leptin action. For instance, it has been suggested that the gut microbiota is associated with leptin resistance by affecting hypothalamic and brainstem neural circuits that regulate feeding and energy balance [139]. Remarkably, the modulation of gut microbiota composition by prebiotics improved leptin sensitivity in diet-induced obese mice and improved glucose tolerance and reduced oxidative stress and low-grade inflammation in ob/ob mice [140]. Moreover, the oral administration of inulin, which is used as a prebiotic to ameliorate glucose and lipid metabolism disorders by modulating the gut microbiota, in ob/ob mice partially reversed the cecal transcriptomic changes in *LEP* gene-related signaling pathways, especially AMPK signaling pathways [141]. Furthermore, other studies found that the administration of *Bacteroides uniformis* CECT 7771 in obese mice reduced body weight gain, as well as plasma cholesterol, triglyceride, and glucose levels, and improved oral glucose tolerance and leptin levels [142]. Overall, these results may suggest that modulation of the gut microbiota could be a novel therapeutic target to target leptin signaling during obesity and obesity-related complications, including CRC.

## 5. Conclusions

At present, obesity and CRC are still major and global concerns for human health, with epidemiological studies revealing their strong and positive correlation. The association of leptin signaling and genetic variation with obesity and CRC is well known, but the genetic component of obesity and CRC is still in focus, whereby mutation gain is a crucial factor enabling the genetic process of CRC. Recent studies have indicated that metabolic changes are able to drive genetic and epigenetic alterations, involving specific pathways and at different time points, thus promoting tumor development. The cumulative and lifetime risk of CRC is considerably increased in cases of a specific hereditary background; however, a considerable percentage of CRCs clustered in families can be ascribed to somatic genomic events, proving the relevance of both the hereditary and the acquired components, as well as the effect of other risk factors. So far, the inconsistent results regarding the association of *LEP* mutations or polymorphisms with obesity and CRC may be due to factors related to genetic and clinical heterogeneity, sample size, population, ethnicity, dietary, lifestyle, age, sex, exposure, or malignancy grade. Therefore, the identification of *LEP* genetic variants associated with human susceptibility to obesity and CRC, as well as those related to their pathogenesis, and specific interactions between genetic and other risk factors not yet fully understood represent critical issues, necessitating wide screening studies.

The present review highlights the need for holistic approaches including genetics, to establish the connections and boundaries of leptin’s involvement in obesity and CRC, as well as their potential link. Such efforts should integrate the microbiome–host interplay by means of advanced metagenomics, metatranscriptomics, metaproteomics, and metabolomics, to assess the leptin genetics–gut microbiome–obesity–CRC relationship. Moreover, the interactions between the genetically acquired and inherited variation in *LEP* related to obesity, the gut microbiome, and CRC still require investigation to provide new insights yielding novel medical approaches. Furthermore, research at different life periods, starting in infancy and childhood with follow-up in adulthood, is needed with a hope to underpin the key factors in preventing or reducing obesity and CRC. Alterations in *LEP*, as well as other genes, are linked to obesity and promote the development of CRC through various signaling pathways. Our understanding of the prospective link between obesity and CRC stems from the potential relationship of *LEP* genetics with the main pathogenic pathways and the molecular mechanisms of these associations, in the context of inconsistent and contradictory data. The understanding of such aspects of leptin biology can help to develop novel therapeutic targets and prevention strategies, resulting in better prognosis and management of these diseases.

## Figures and Tables

**Figure 1 ijms-23-04713-f001:**
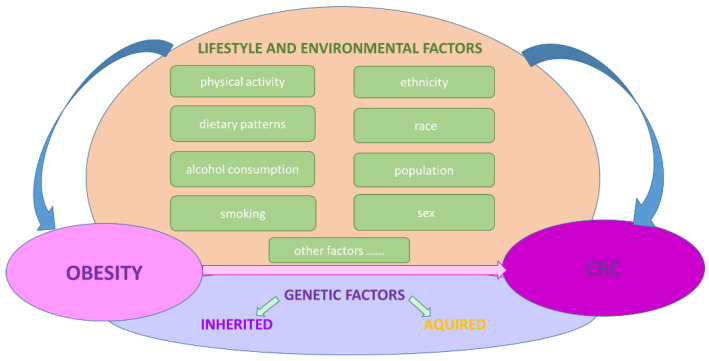
Factors influencing obesity and CRC risk.

**Figure 2 ijms-23-04713-f002:**
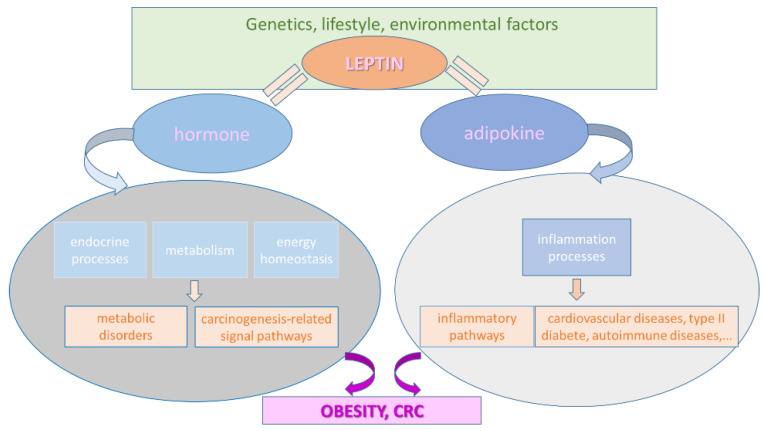
Leptin’s role in obesity and CRC.

**Figure 3 ijms-23-04713-f003:**
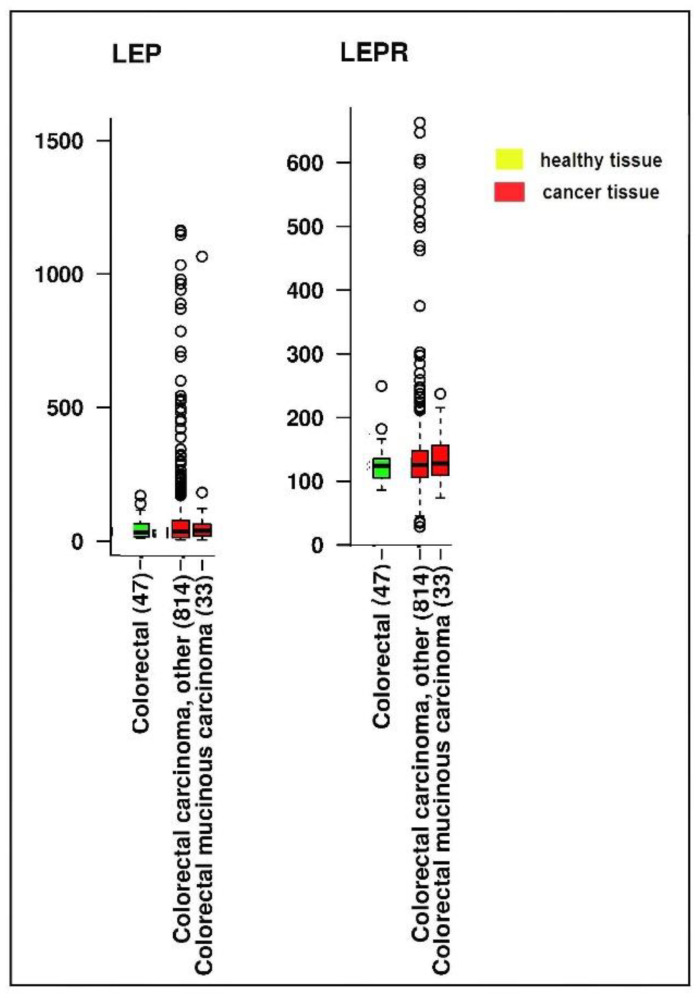
*LEP* and *LEPR* gene expression levels in healthy and in various colorectal carcinoma tissues. Data were retrieved from the online IST database (https://ist.medisapiens.com/, accessed on 24 June 2021).

**Figure 4 ijms-23-04713-f004:**
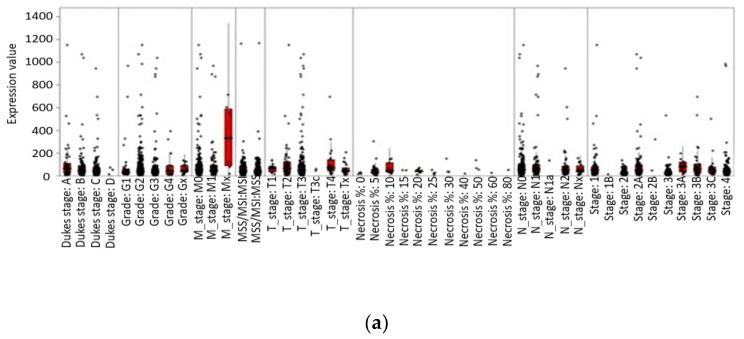
*LEP* and *LEPR* gene expression association with colorectal cancer phenotypes. Data represent the expression values of *LEP* (**a**) and *LEPR* (**b**) in colorectal cancer samples with several types of clinical data, within each cancer dataset. Each sample is represented by a single datapoint. Each type of clinical data exists as a column within a separate segment of the phenoplot. Expression values are also presented as red box plots. Boxes represent samples with distinct phenotypic values as black dots in the clinical data segments. A box is only shown if there are more than five distinct phenotypic values. Data were retrieved from the online IST database (https://ist.medisapiens.com/, accessed on 24 June 2021).

**Figure 5 ijms-23-04713-f005:**
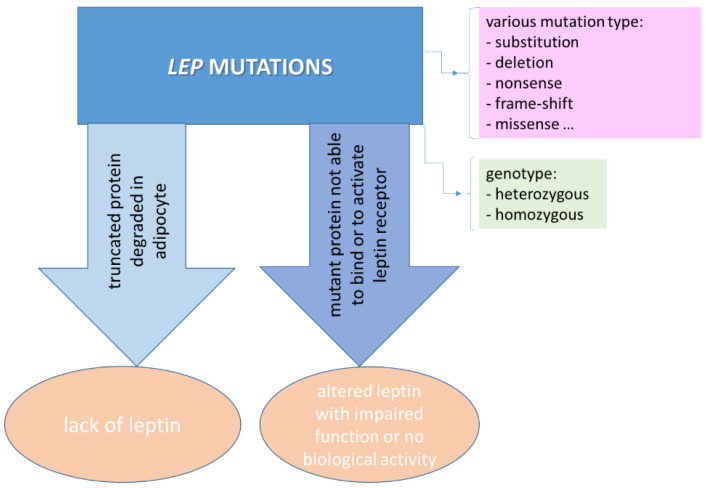
Effect of *LEP* mutations on leptin expression in obesity and CRC.

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
