# Peer review of "Leptin Signaling in Obesity and Colorectal Cancer"

_ijms, 2022, doi:10.3390/ijms23094713_

Round 1

Reviewer 1 Report

In general, this work is really interesting and well written. My comments aim to increase the scientific soundness and clarity of it.

  1. The topic of this article is not new and has been presented several times before. There are many more comprehensive articles dealing with the same subjects.

See:

a) Yang R, Barouch LA. Leptin signaling and obesity: cardiovascular consequences. Circ Res. 2007 Sep 14;101(6):545-59. doi: 10.1161/CIRCRESAHA.107.156596. PMID: 17872473.

b) Uddin S, Hussain AR, Khan OS, Al-Kuraya KS. Role of dysregulated expression of leptin and leptin receptors in colorectal carcinogenesis. Tumour Biol. 2014 Feb;35(2):871-9. doi: 10.1007/s13277-013-1166-4. Epub 2013 Sep 7. PMID: 24014051.

c) Khodamoradi K, Khosravizadeh Z, Seetharam D, Mallepalli S, Farber N, Arora H. The role of leptin and low testosterone in obesity. Int J Impot Res. 2022 Jan 31. doi: 10.1038/s41443-022-00534-y. Epub ahead of print. Erratum in: Int J Impot Res. 2022 Mar 8;: PMID: 35102263.

d) Picó C, Palou M, Pomar CA, Rodríguez AM, Palou A. Leptin as a key regulator of the adipose organ. Rev Endocr Metab Disord. 2022 Feb;23(1):13-30. doi: 10.1007/s11154-021-09687-5. Epub 2021 Sep 14. PMID: 34523036; PMCID: PMC8873071.

To make this article more interesting to a reader I suggest the authors to precisely and clearly define the target question of this review.

2.      Chapter 1 is too long. I understand it was intended as an introduction, but as it stands it contains too much superflous information.

  1. The term “gut” should be more specified. There will be substantial differences between small and large intestines.
  2. Figure 3-5. What LEP stands for ?
  3. The abbreviation LEP is also used several times in the main text. If it means leptin, authors should consistently abbreviate the rest of the names. If it stands for the name of gene than it should be written in italics.
  4. Conclusion does not contain any future perspectives in the field.

Reviewer 2 Report

The correlation with red non-processed meat is not proven; you should underline processed....

The authors state that obesity is characterized by a genetic component, but the fact that it is frequent in consanguine could be due to common incorrect habits and not to a genetic component

The link with the microbiota is very hypothetical, in particular, it is much more plausible that inflammation due to obesity (partly mediated by leptin, but not only, as there are many inflammatory cytokines), alters intestinal permeability and functionality, making the intestine itself more prone to oncogenesis

Therefore the work should be revised by limiting or eliminating the part concerning the microbiota, underlining the part relating to a possible mechanism concerning leptin which in any case cannot be univocal

Round 2

Reviewer 2 Report

I think that the authors made substantial changes so the manuscript could be published.

Author Response

The authors would like to thank you for your time and your positive feedback, appreciated!